# Synthesizing topological structures containing RNA

Di Liu[1], Yaming Shao[2], Gang Chen[1], Yuk-Ching Tse-Dinh[3], Joseph A. Piccirilli[1,2] & Yossi Weizmann[1]

Though knotting and entanglement have been observed in DNA and proteins, their existence in RNA remains an enigma. Synthetic RNA topological structures are significant for understanding the physical and biological properties pertaining to RNA topology, and these properties in turn could facilitate identifying naturally occurring topologically nontrivial RNA molecules. Here we show that topological structures containing single-stranded RNA (ssRNA) free of strong base pairing interactions can be created either by configuring RNA–DNA hybrid four-way junctions or by template-directed synthesis with a single-stranded DNA (ssDNA) topological structure. By using a constructed ssRNA knot as a highly sensitive topological probe, we find that *Escherichia coli* DNA topoisomerase I has low RNA topoisomerase activity and that the R173A point mutation abolishes the unknotting activity for ssRNA, but not for ssDNA. Furthermore, we discover the topological inhibition of reverse transcription (RT) and obtain different RT–PCR patterns for an ssRNA knot and circle of the same sequence.

[1] Department of Chemistry, The University of Chicago, Chicago, Illinois 60637, USA. [2] Department of Biochemistry and Molecular Biology, The University of Chicago, Chicago, Illinois 60637, USA. [3] Department of Chemistry and Biochemistry, Biomolecular Sciences Institute, Florida International University, Miami, Florida 33199, USA. Correspondence and requests for materials should be addressed to Y.W. (email: yweizmann@uchicago.edu).

Knotting and entanglement are not only common macroscopic phenomena, but also present at the molecular level, via either random statistical threading[1,2] or elegant rational designs[3–5]. The occurrence of molecular topology is also frequent in biological context[6], and two of the most important biological macromolecules, DNA[7,8] and proteins[9,10], have been found to adopt nontrivial topologies. Whereas the functional implications of knotted proteins still remain unclear, DNA topology is a prominent and fundamental theme in modern biology, and largely defines the structural, biological and functional principles of DNA and most DNA-processing enzymes[8,11]. In fact, the biological importance of DNA topology can partly be reflected by the diversity of DNA topoisomerases (DNA Topos)[12–14], which are enzymes evolved to solve the topological problems of DNA.

An interesting question naturally arises regarding the importance of RNA topology. Although the recent systematic screening of the Protein Data Bank indicated the absence of genuine linear knots in known RNA structures[15,16], it is yet premature to disclaim the existence of naturally occurring knotted (linear or closed) RNA structures. On the one hand, the set of solved RNA structures are not representative of all RNA molecules. There are many more RNAs with unknown structures, and RNAs that remain to be identified. It is likely that knotted RNA structures can be discovered as more and more RNA structures are solved. On the other hand, RNA pseudoknots with two sufficiently long (at least around 11 bp) helices can adopt knotted conformation and several likely knot-forming candidates have been suggested[16] based on their sequences from an RNA pseudoknot database[17]. Instead of passively waiting for RNA topological structures to emerge through the accumulation of structural data, RNA topologies can be created and investigated through synthesis. Importantly, synthetic RNA topological structures can help us understand the physical and biological properties associated with RNA topology. Based on these properties, new tools and methods to identify the naturally occurring RNA topological structures can be developed.

Synthetic DNA topology is an active field, where various nanoscale DNA topological structures have been constructed and functionalized[18–20]. However, synthetic RNA topology has received far less attention. So far, the only method to access it was described by Seeman and co-workers[21]. As a sequel to a series of their remarkable studies on single-stranded DNA (ssDNA) topologies[22–25], a single-stranded RNA (ssRNA) trefoil knot was constructed by utilizing the intrinsic topological properties of an RNA duplex[21]. The realization of synthetic RNA topology essentially led to the discovery of the first enzyme with RNA topoisomerase (RNA Topo) activity—*Escherichia coli* DNA Topo III (ref. 21). Based on this, RNA Topo activity has also been recently found in other Type IA DNA Topos[26,27], including the human DNA Topo 3β (ref 26), which is crucial to neurodevelopment. RNA topology and RNA Topos, similar to their DNA counterparts, have the potential to transform our understanding of fundamental RNA biology.

Here we demonstrate that synthetic RNA topologies can be accessed either by configuring the RNA–DNA hybrid four-way junction (4WJ), or by template-directed synthesis using a ssDNA topological structure. The resulting RNA topological structures are free of strong base-pairing interactions, enabling the RNA Topo activity study of *E. coli* DNA Topo I and the discovery of topological inhibition of reverse transcription (RT). We expect our work on synthetic RNA topological structures will stimulate research in the essentially unexplored area of RNA topology and RNA topoisomerase.

## Results

**Strategies for the construction of ssRNA topological structures.** The first ssRNA knot was constructed by Seeman's helix-based method[21], the principle of which is that a half-turn (5 or 6 bp) of RNA duplex generates a node (Fig. 1a,b). Recently, we have expanded the spectrum of synthetic DNA topologies with a versatile method based on the stacked X structure of DNA 4WJ, in which the two helical strands (continuous along the stacked helices) are held by the two crossover strands (exchanged at the junction) to form a node for topological construction[28]. Compared with the previous helix-based method, this junction-based method has three major advantages. First, the resulting topological constructs contain no intrinsic strong base pairings. Second, the method enables the convenient generation of both positive and negative nodes. Finally, the method circumvents undesired braiding of the ssDNA linkers that is frequently encountered in the helix-based method[29]. This junction-based method can be further developed for synthetic RNA topologies by using the RNA–DNA hybrid 4WJ (Fig. 1c), which contains RNA helical strands and DNA crossover strands. Thus, RNA topological structures can be generated by folding RNA scaffolds into hybrid 4WJs with DNA staples. Figure 1d illustrates the construction of an RNA trefoil knot using the helix-based method. Alternatively, DNA-templated synthesis can be used to construct an RNA topology, that is, a pre-prepared ssDNA topological structure can be used as a template to synthesize the corresponding RNA structure (Fig. 1e).

**Right- and left-handed RNA trefoil knots with the same sequence.** To produce closed RNA topological structures, the linear RNA strands can be ligated enzymatically. However, RNA strands synthesized by *in vitro* runoff transcription, which is the most convenient and economic method to produce long RNA strands, contain both 5′- and 3′-heterogeneities. Therefore, their enzymatic ligation poses a significant challenge. To solve these problems, a 5′-end hammerhead ribozyme and a 3′-end hepatitis delta virus ribozyme[30] were designed in each RNA transcript to undergo self-cleavage to produce uniform ends (Fig. 2a). T4 polynucleotide kinase is then used to add a phosphate group to the 5′-hydroxyl group (in the presence of ATP) and remove the 2′,3′-cyclic phosphate[31] of self-cleaved RNA transcripts. The resulting RNA molecules contain both ends proper for ligation and can serve as the scaffolds for topological construction.

The RNA–DNA hybrid 4WJs presumably have structural properties similar to that of DNA 4WJs due to the fact that the former have been utilized in the construction of various RNA–DNA hybrid nanostructures[32,33], within which the 4WJs are composed of helical strands of RNA and crossover strands of DNA. Therefore, the design principles of DNA nanostructures based on DNA 4WJs also hold for those containing the RNA–DNA hybrid 4WJs. The tensegrity triangle[34], a structural motif containing three 4WJs, is a case in point and it is utilized extensively in this work for topological construction. The number of base pairs between the 4WJs dictates the tensegrity triangle to be either right- or left-handed[35], which results in positive or negative nodes, respectively. The construction of topoisomers of both right- and left-handed trefoil knots can be achieved by folding the same RNA scaffolds into different tensegrity triangles with different sets of DNA staples (Fig. 2b). After geometrical analysis according to previous work[28,35] (see also Supplementary Table 1), we used a 17-bp-edged tensegrity triangle for the right-handed RNA trefoil knot $(3_1^+)$, **TKj(+)** and 14-bp-edged for the left-handed trefoil knot $(3_1^-)$, **TKj(−)**. Notably, for the helix-based method, generating positive nodes

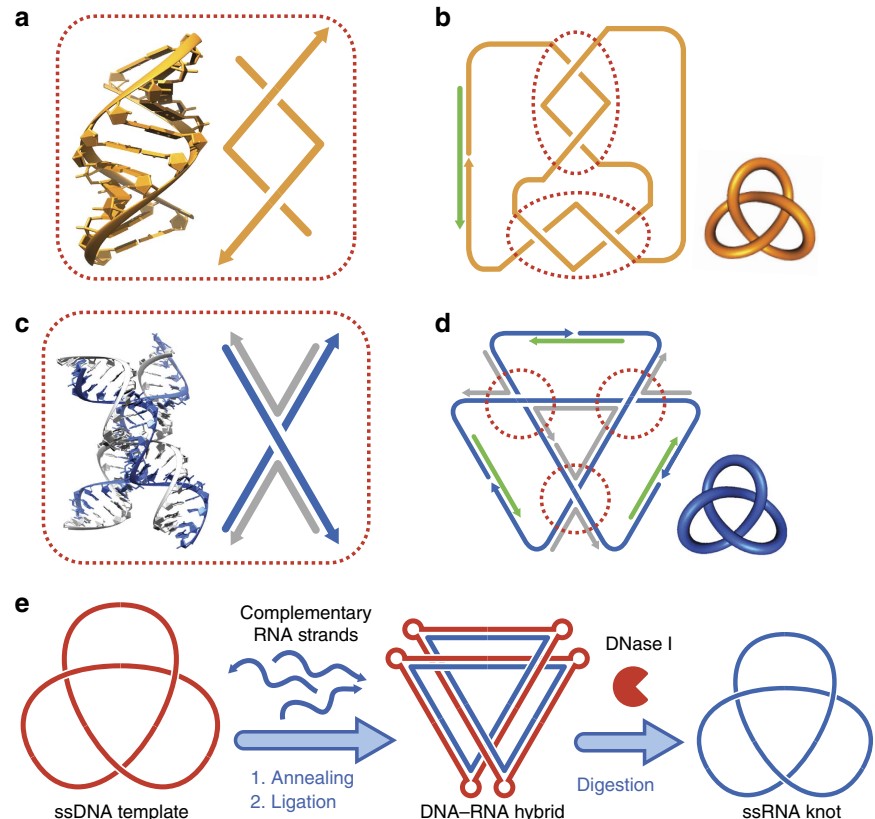

**Figure 1 | Strategies of constructing ssRNA topological structures. (a,b)** Seeman's method of using A-form RNA helix to generate ssRNA topological structures[21]. In **a**, one turn of an A-form RNA helix is shown with the helical and schematic representations. The two component strands form two negative nodes within this one-turn helix. In **b**, a strand of ssRNA (orange) is designed to contain alternating complementary pairing segments to form two one-turn A-form helices, and a trefoil knot is formed after enzymatic ligation aided by a DNA splint (green). However, topological structures constructed in this way contain very strong intrastrand base pairings. **(c,d)** Junction-based method to generate ssRNA topological structures. In **c**, a 4WJ is formed with two RNA strands (blue) as the helical strands and two DNA strands (grey) as the crossover strands, and is shown with the helical and schematic representations. The two RNA helical strands generate a node. In **d**, the assembly complex for the trefoil knot is formed, where the RNA scaffolds (blue) are threaded into the targeted topology by DNA staples (grey) and linked end to end by DNA splints (green). After ligation and subsequent removal of the DNA staples and splints, a ssRNA knot free of strong intrastrand base pairings is generated. **(e)** The ssDNA trefoil knot can be used as a template for the construction of ssRNA trefoil knot. The ssDNA knot template (red) is pre-prepared and annealed with the complementary RNA strands (blue). After ligation, the DNA–RNA hybrid knot is formed. The ds DNA–RNA hybrid is more rigid than single-stranded structure, the careful design of curvature (by adding bulges) and torsion (by adjusting the length of hybrid helix) is necessary. The hybrid knot can then be subjected to DNase I digestion to obtain the ssRNA knot.

would demand the formation of left-handed Z-form RNA, which requires harsher conditions (higher salt concentration and higher temperature)[36] than the formation of Z-form DNA. In contrast, our junction-based method provides a more convenient way to generate the positive nodes because the handedness of the nodes is controlled by applying geometric constraints to 4WJs, but not by the formation of nonstandard duplex structures.

Experimentally, three 76-nt ssRNA scaffolds were annealed with three DNA splints and either set of four DNA staples. In the resulting assembled complex, the ssRNA scaffolds are folded by the staples to form three 4WJs (configured within the tensegrity triangle) and joined end to end by the splints (Fig. 2b). Subsequent ligation seals the nicks in between the scaffolds and thereby fixes the topology. T4 RNA ligase 2 was used for the DNA-splinted RNA ligation because lower enzyme concentration is needed compared to T4 DNA ligase (which ligates RNA less efficiently and only catalyses approximately stoichiometric ligation). All the assisting DNA strands (staples and splints) are dissociated on purification with denaturing polyacrylamide gel electrophoresis (dPAGE), and consequently relaxed 228-nt ssRNA topological constructs are generated without strong

intrinsic intramolecular base pairings. $TK_j(+)$ and $TK_j(-)$ were obtained in yields of 13% and 28%, respectively (Supplementary Fig. 1), comparable to the previous synthesis of DNA trefoil knots of similar size[28]. Figure 2c shows the purified RNA trefoil knots analysed by dPAGE: lane 1 for $TK_j(+)$, lane 3 for $TK_j(-)$, along with the circular ($C_j$, lane 5) and linear ($L_j$, lane 7) references. Similar to the previous results with DNA topological structures[28], $TK_j(+)$ and $TK_j(-)$ have almost identical electrophoretic mobility, which is higher than that of $C_j$. The resistance to digestion with RNase R proves the closed structure for knots (lanes 2 and 4) and circle (lane 6), but not for the linear species (lane 8).

**Hybrid Borromean rings (BR).** To further demonstrate the high-level complexity of topological structures we can achieve with RNA, we create a hybrid Borromean rings (BR) $\left(6_2^3\right)$ structure containing one ssRNA and two ssDNA rings. The defining feature of this famous topology is that the whole assembly of three rings falls apart on the scission of any one ring. This requires an equal number of positive and negative nodes

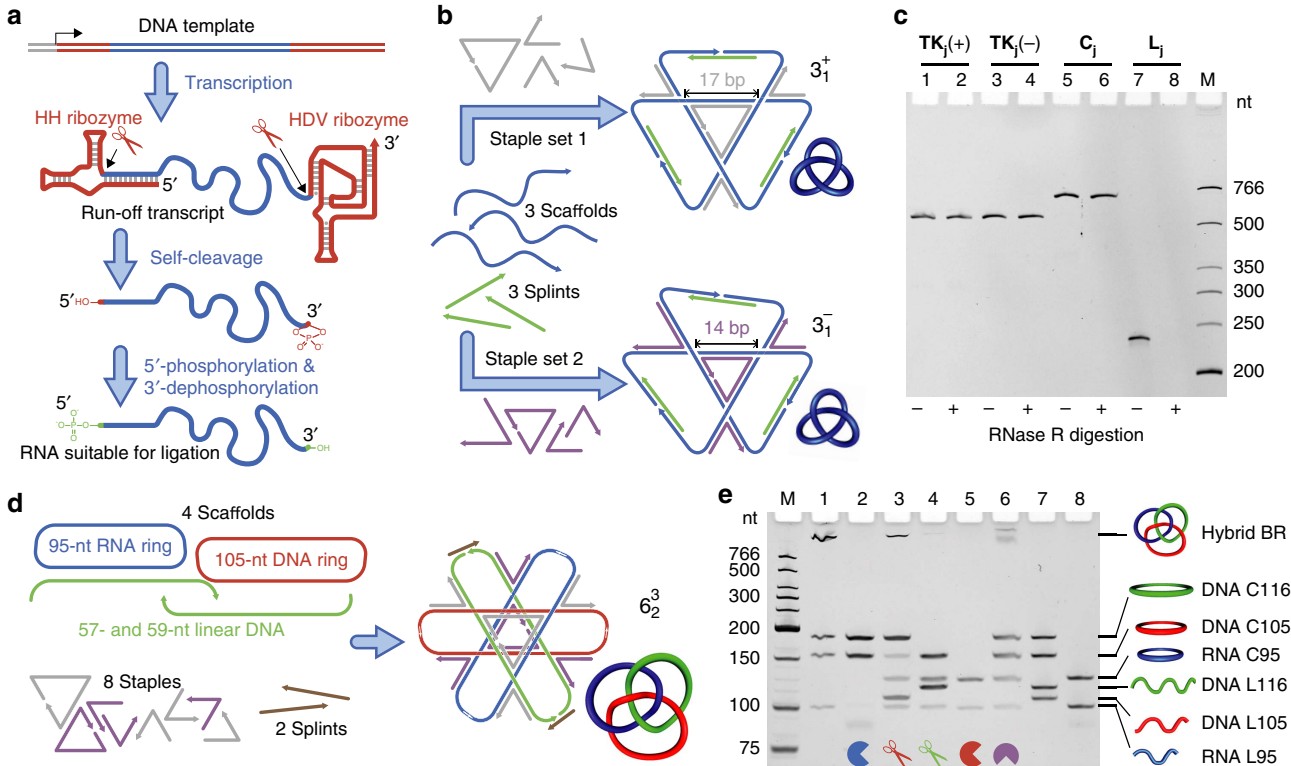

**Figure 2 | Constructing ssRNA topological structures with the junction-based method.** (**a**) Preparation of the ssRNA strand with uniform ends and proper end groups for the DNA-splinted RNA ligation. (**b**) Both the positive ($\mathbf{TK_j}(+)$) and negative ($\mathbf{TK_j}(-)$) RNA trefoil knots of the same sequence are constructed by configuring tensegrity triangles with different handedness. The same scaffolds (blue) are threaded by different staple sets (grey or purple) to form a 17-bp-edged right-handed tensegrity triangle for the positive trefoil knot, or a 14-bp-edged left-handed tensegrity triangle for the negative one, respectively. Each topology is designated by the Alexander–Briggs notation $n_i$ or $n_i^C$, where $n$ is the minimal number of nodes, $C$ is the number of components (for links), and $i$ distinguishes different topologies with the same $n$ and $C$. (**c**) dPAGE analysis of $\mathbf{TK_j}(+)$ (lanes 1 and 2), $\mathbf{TK_j}(-)$ (lanes 3 and 4), and their circular ($\mathbf{C_j}$, lanes 5 and 6) and linear ($\mathbf{L_j}$, lanes 7 and 8) counterparts. Lanes 2, 4, 6 and 8 contain samples digested by RNase R. (**d**) The assembly complex for the hybrid BR contains tensegrity triangles of both handedness to generate three positive nodes plus three negative nodes. (**e**) Topological analyses of the hybrid BR. Lane 1, gel-purified BR; lanes 2–6, BR treated by RNase H, Nt.AlwI (for cleaving the red ring), Nt.BspQI (for cleaving the green ring), DNase I, and *E. coli* DNA Topo I; lanes 7 and 8, DNA and RNA references of the three individual components. During the purification of BR, the breaking down of the 95-nt circular RNA component is unavoidable, and a portion of BR falls apart as a result. The treatment of BR by RNase H, Nt.AlwI and Nt.BspQI is conducted in the presence of an assisting DNA strand complementary to the corresponding ssRNA or ssDNA ring. LX and CX represent X-nt linear and circular species, respectively. In all the gels, lane M contains the DNA size markers.

to ensure that no two rings are interlocked. In our previous construction of ssDNA BR[28], two tensegrity triangles with different handedness were designed in the assembly complex to meet this requirement. Similarly, in the assembly complex for this hybrid BR, a 95-nt circular RNA, a 105-nt circular DNA and two linear DNA (57- and 59-nt, respectively) scaffolds are folded into a 17-edged right-handed tensegrity triangle for the three positive nodes and a 14-edged left-handed one for the three negative nodes (Fig. 2d). The two linear DNA scaffolds (precursors for the 116-nt DNA ring) are joined by two splints and the hybrid BR structure is formed after ligation by T4 DNA ligase (Supplementary Fig. 2). To conclusively prove the topology of this hybrid BR, each DNA ring is installed with a unique restriction site. As shown in Fig. 2e, the hybrid BR is disassembled by the cleavage of the ssRNA ring by RNase H (lane 2), or either ssDNA ring by the corresponding nickase (lanes 3 and 4).

In the field of chemical topology, molecular BR is a well-known challenging target[37], attracting researchers from different disciplines to develop various novel strategies to realize it[25,28,38–41]. Here we further extend the chemical diversity of this topological target by creating this hybrid BR structure,

which is the first BR molecule to contain component rings of different materials. To the best of our knowledge, it is also the first topological structure composed of both DNA and RNA. Additionally, the successful construction of a topological target as complex as BR reasonably implies that our junction-based method would provide access to the ssRNA or ssRNA–ssDNA hybrid versions of any of our previous topological targets realized with ssDNA[28].

**ssRNA knot constructed via DNA-templated synthesis.** Synthetic ssDNA topological structures constructed previously, in principle, can direct the synthesis of ssRNA structures of the same topology (Fig. 1c). The intermediates would be the double-stranded (ds) RNA–DNA hybrid structures; however, the conversion of the structures from ssDNA to ds version is not always straightforward. This is especially true for small ds knots, which are more difficult to synthesize than other ds topological structures that contain only rings (such as rotaxanes[5] or catenanes[6]). Because ds nucleic acid structure is more rigid and adopts a better-defined geometry compared to ssDNA or ssRNA, constructing ds knot necessitates the careful design of curvature and torsion in 3D space[28]: (1) the total curvature

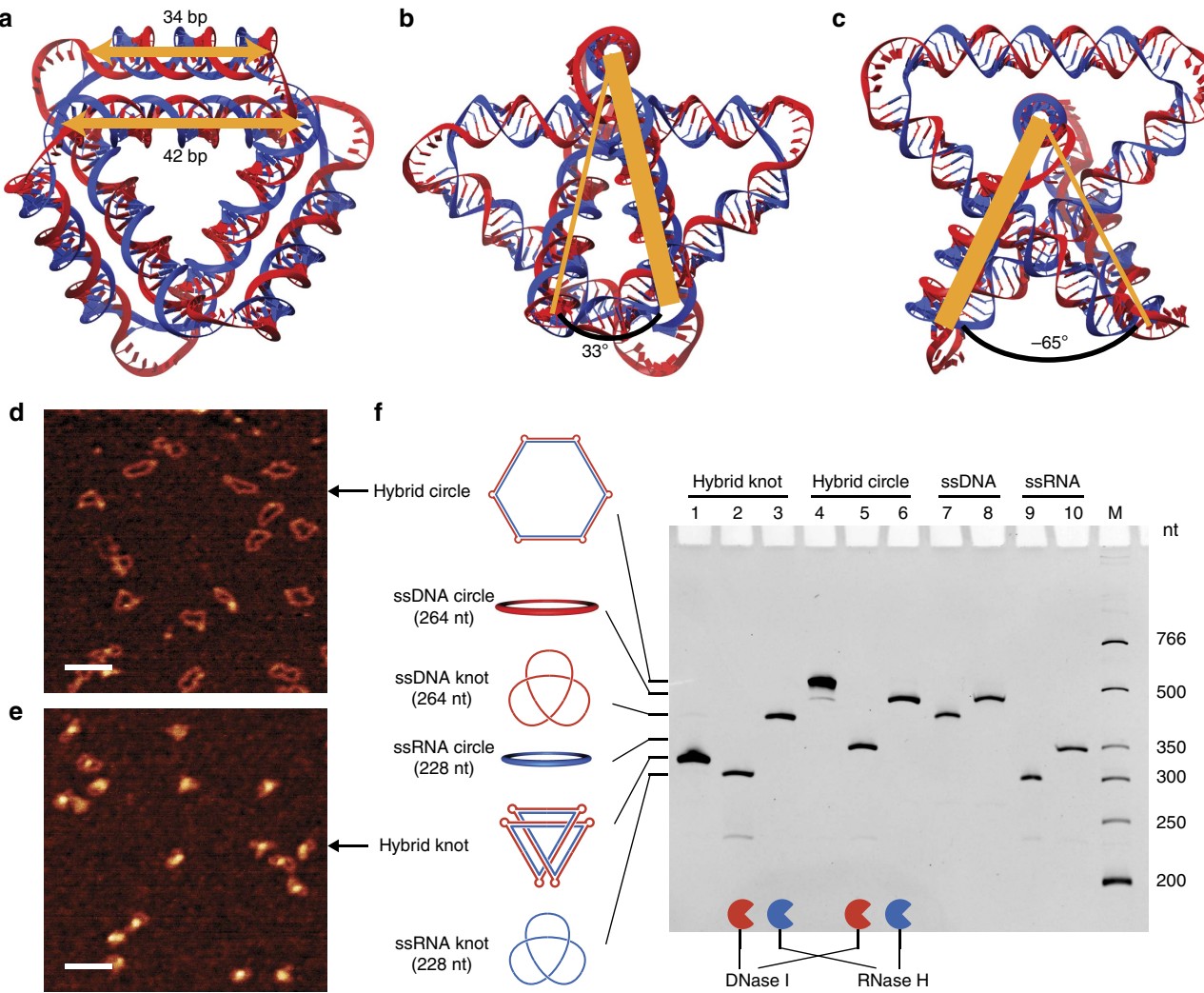

**Figure 3 | Design and construction of a ds DNA–RNA hybrid knot.** (**a–c**) Three views of 3D helical model of the ds DNA–RNA hybrid knot: along the threefold rotation axis (**a**), the axis of the outer (**b**) and inner (**c**) helices. The template DNA strand is shown in red and the complementary RNA strand in blue. Also see Supplementary Fig. 3 for stereo views. (**d**,**e**) AFM images for the ds hybrid circle (**d**) and knot (**e**). Scale bar, 50 nm. (**f**) Nuclease digestion confirming the formation of the hybrid knot. Lanes 1–3, purified ds hybrid knot undigested (lane 1), digested by DNase I (lane 2) or by RNase H (lane 3). Lanes 4–6, purified ds hybrid circle undigested (lane 4), digested by DNase I (lane 5) or by RNase H (lane 6). Lanes 7 and 8, ssDNA references of the ssDNA template knot and circle. Lanes 9 and 10, ssRNA references (**TK_j**(+) and **C_j**), which are of the same size with the RNA strand in the ds hybrid structures but a different sequence.

of a knot should be larger than $4\pi$ (more than two times that required for a circle) according to the Fary–Milnor theorem; (2) the torsion of a knotted space curve should not be zero everywhere because a knot cannot be flattened onto the plane.

To construct the ds RNA–DNA hybrid knot, a 264-nt ssDNA-positive trefoil knot was pre-prepared (using the junction-based method) to serve as the template, and three 76-nt complementary RNA strands were then annealed onto the DNA template and ligated to form a 228-nt RNA strand (Supplementary Fig. 4). Six poly(dT)$_6$ bulges are designed in the DNA strand of the hybrid to provide six curving points, and they connect three outer 34-bp helices (generating a 33° torsion angle) and three inner 42-bp ones (generating a −65° torsion angle) into the closed ds knot structure (Fig. 3a–c and Supplementary Fig. 3). If the sequence details are neglected, this ds knot is a *C3*-symmetry molecule. Using ssDNA circle as the template, we also prepared a ds hybrid circle as a topoisomeric reference. The atomic force microscopy (AFM) images of the ds hybrid circle (Fig. 3d) and knot (Fig. 3e) reveal the obvious structural differences

between them, and the ds hybrid knot adopts a more compact structure with strand crossings (reflected by the higher bumps in the AFM image). Furthermore, these ds hybrid structures were digested by nucleases and subsequently analysed by dPAGE (Fig. 3f). The highly compact structure of the hybrid knot is again reflected by the very high electrophoretic mobility (lane 1), which exceeds the hybrid circle (lane 4) and the ssDNA template (lane 7). DNase I digests the DNA strand of either hybrid structure and, as expected, the ssRNA knot (lane 2) or circle (lane 5) is released. The ssDNA templates are recovered by RNase H digestion of the hybrid structures (lanes 3 and 6). Not only does our approach provide the alternative route of DNA-templated synthesis for the construction of ssRNA topological structures, but also demonstrates the first realization of a ds RNA–DNA hybrid topological structure. This kind of ds hybrid structure, with more rigid and better-defined 3D structure, may find other potential applications in nanobiotechnology and nanofabrication. Additionally, the principle of using a knotted DNA template to guide the topology of RNA in the current

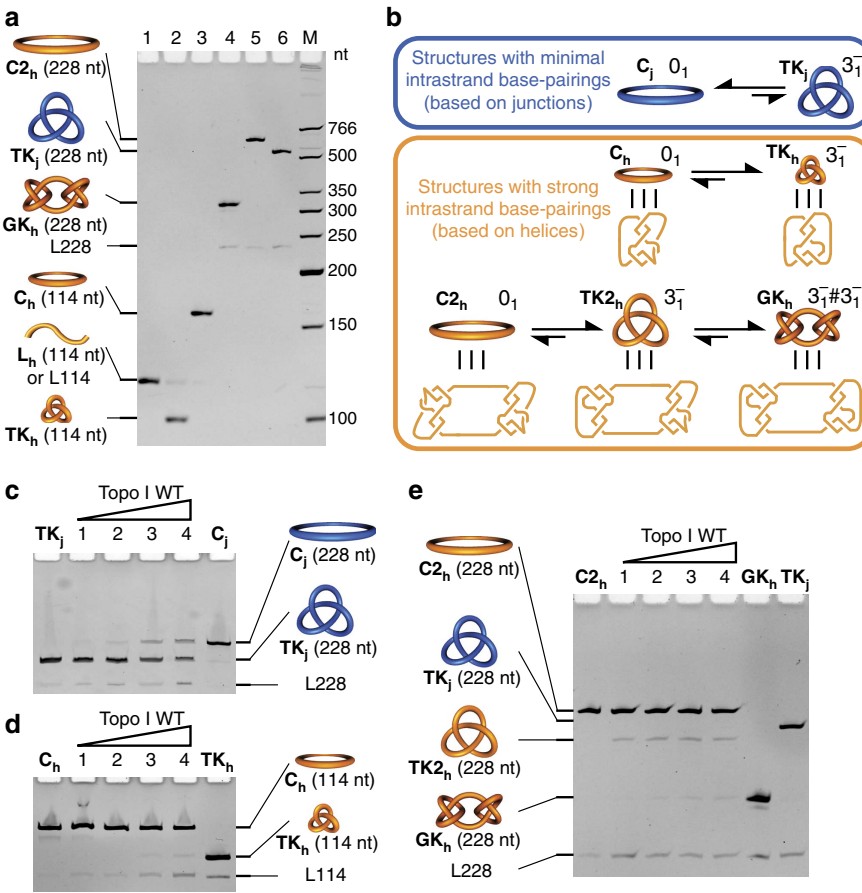

**Figure 4 | Comparing probes for RNA topoisomerase (RNA Topo) activity.** (**a**) dPAGE analyses of various structures: monomeric linear (**Lh**, lane 1), trefoil knot (**TKh**, lane 2), and circular (**Ch**, lane 3) RNA molecules with the sequence for helix-based topological structures; dimeric granny knot (**GKh**, lane 4), and circular (**C2h**, lane 5) species with the same sequence; and the junction-based negative trefoil knot (**TKj**, lane 6). (**b**) Hypothetical conversions of junction-based (between **Cj** and **TKj**) and helix-based (between **Ch** and **TKh**, and between **C2h**, **TK2h** and **GKh**) topological structures under 'ideal' RNA Topo condition, when the RNA strand-passage event can freely take place. (**c–e**) Topological relaxations of **TKj** (**c**), **Ch** (**d**) and **C2h** (**e**) catalysed by increasing concentrations of WT *E. coli* DNA Topo I (30 min incubation at 37 °C). In **c**,**e**, the RNA probe substrates (**TKj** or **C2h**) were 80 nM and Topo I in lanes 1–4 was 40, 80, 160 and 320 nM. In **d**, the RNA probe substrate (**Ch**) was 160 nM and Topo I in lanes 1–4 was 80, 160, 320 and 640 nM. In **e**, **C2h** is relaxed to the trefoil knot **TK2h** and to **GKh** after one and two strand-passage events, respectively. All lanes contain the linear break-down products of the closed RNA structures and they are annotated by LX, in which X represents X-mer.

research reveals the possibility of synthesizing topological structures of other non-nucleic acid materials with the more general DNA-templated synthesis[42].

**Probing RNA topoisomerase activity.** One of the most significant discoveries resulting from synthetic RNA topology is the identification of *E. coli* DNA Topo III as the first enzyme possessing RNA Topo activity[21]. More recently, several other DNA Topos were also reported to have RNA Topo activity[26,27]. Unlike DNA Topos, the research on RNA Topos is belated and rare, mainly due to the lack of proper RNA Topo probes. The ssRNA knot prepared with our junction-based method does not contain strong base pairings, and here we show that it serves as a more sensitive probe for RNA Topo activity compared with the previous helix-based probe[21]. We tested *E. coli* DNA Topo I (denoted as Topo I afterwards for clarity) for the RNA Topo assay, and in contrast to previous report[21], we find that Topo I indeed possesses RNA Topo activity. (While we were preparing the paper, the RNA Topo activity of Topo I was independently reported[27].)

We used our ssRNA negative trefoil knot, **TKj**( − ) (simplified as **TKj** afterwards), as the topological probe, which would be

converted to the ssRNA circle, **Cj**, in the presence of an RNA Topo activity. Additionally, four RNA structures containing strong base pairings were constructed from the same 114-nt linear RNA strand, **Lh**, according to Seeman's helix-based method[21] with minor modifications. By adding different sets of assisting DNA strands in the synthesis, the monomeric trefoil knot, **TKh**, and circle, **Ch**, and the dimeric Granny knot ($3_1^- \# 3_1^-$, a complex knot), **GKh**, and circle, **C2h**, were prepared (Supplementary Figs 5 and 6). Figure 4a shows dPAGE analyses of these structures and the electrophoretic mobility is a function of both size and topology. Figure 4b illustrates the topological conversion of these structures under 'ideal' RNA Topo conditions, that is, when the RNA strand-passage events occur freely and lead to the most thermodynamically stable topoisomers. The most stable topoisomer for the structures free of strong base pairings (with junction-based method) is that of a simpler topology (**Cj**). In contrast, the most stable topoisomers of the helix-based structures are those favouring maximum base pairings and consequently are those of more complex topology (**TKh** and **GKh** respectively). Therefore, **Ch**, and **C2h** were used as the helix-based RNA Topo probes to be compared with the junction-based **TKj** (also

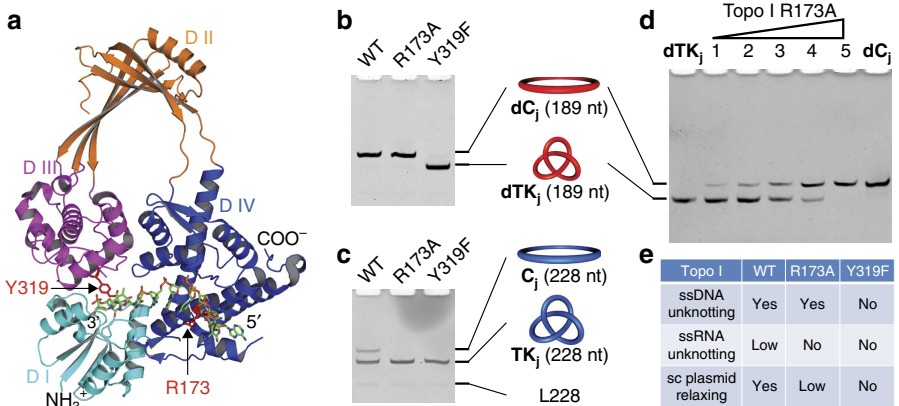

**Figure 5 | Activity of Topo I mutants against different substrates.** (**a**) Crystal structure of the Topo I ssDNA covalent complex (Protein Data Bank ID: 3PX7)[43]. The four domains (D I–IV) of the enzyme are shown in different colours, with the two mutated residues in mutant study, R173 and Y319, highlighted in red. (**b,c**) Investigating the ssDNA- and ssRNA-unknotting activity of WT Topo I and two mutants using ssDNA knot, **dTK$_j$** (**b**) and ssRNA knot, **TK$_j$** (**c**) as probes. Probe substrates were 80 nM, treated by 320 nM of proteins (30 min incubation at 37 °C). (**d**) Topological relaxations of **dTK$_j$** (80 nM) catalysed by increasing concentrations of Topo I R173A mutant (30 min incubation at 37 °C). The enzyme in lanes 1–5 was 10, 20, 40, 80 and 160 nM. (**e**) Summary of activities of the WT Topo I and two mutants tested in this work against different substrates. The sc plasmid relaxation activities were taken from previous work[44].

see Supplementary Fig. 7 for more explanation of why **C$_h$**, but not **TK$_h$**, is used as the helix-based probe).

Figure 4c shows the increasing conversion of **TK$_j$** to **C$_j$** catalysed by increasing concentrations of wild-type (WT) Topo I. Though Topo I has RNA Topo activity, this activity is relatively low and ∼28–35% of conversion was observed when the molar ratio of Topo I to RNA is 4:1 (lane 4). Based on our previous result that Topo I catalyses the fast approximately stoichiometric unknotting of ssDNA knot within 30 min[28], the RNA Topo activity of Topo I is estimated to be ∼1/15–1/12 of the DNA Topo activity. Comparing with **TK$_j$**, **C$_h$** is a much less sensitive RNA Topo probe (Fig. 4d). Only ∼3% conversion was observed when the molar ratio of Topo I to RNA is 4:1 (lane 4). Figure 4e shows the RNA Topo assay using probe **C2$_h$**. Though **C2$_h$** is more sensitive than **C$_h$**, probably due to the more severe topological stress, it is still not as sensitive as **TK$_j$** at high enzyme-to-RNA ratios. As expected, there are two products of the topological conversion of **C2$_h$**, the larger-amount intermediate trefoil knot **TK2$_h$** (after one strand-passage event), and the smaller-amount final **GK$_h$** (after two strand-passage events). We also find that **TK2$_h$** migrates slightly faster than **TK$_j$**, though both are of the same size and topology, probably due to the different sequences or the formation of transient base pairings within **TK2$_h$** even during migration in the dPAGE.

The observation that **C$_h$** and **C2$_h$** (helix-based) are not as sensitive probes as **TK$_j$** (junction-based) can be explained by both the Topo I binding to ssRNA and its low RNA Topo activity. Topo I binding is expected to inhibit the formation of base pairings, countering the thermodynamic driving force for the topological conversion illustrated in Fig. 4b for **C$_h$** and **C2$_h$**. This problem is further exaggerated due to the low RNA Topo activity, which necessitates a higher concentration of Topo I and ultimately leads to even more severe protein binding. The low sensitivity of the helix-based probe may account for the previous failure of finding RNA Topo activity of Topo I (ref. 21). Besides, better sensitivity would circumvent the inconvenience associated with the use of autoradiography with $^{32}$P-labelled RNA as described in the recent study[27].

**Substrate-specificity study of Topo I mutants.** In the crystal structure of the Topo I ssDNA covalent complex[43], domain IV of the enzyme provides several contacts with the DNA substrate and is important for the binding and recognition of the substrate

(Fig. 5a). A key residue is R173, which interacts with the −4 position cytosine base via hydrogen bonding. Previous studies demonstrated that the R173A point mutation displays an ∼100-fold decrease in the relaxation activity of supercoiled (sc) plasmid DNA[44] and completely loses the ability to relax the helix-based ssRNA probe[27]. With our junction-based probes of both ssDNA (a previously prepared ssDNA trefoil knot, **dTK$_j$**)[28] and ssRNA (**TK$_j$**), the R173A mutant was investigated regarding its relaxation activities of ssDNA and ssRNA. Interestingly, we could still detect the unknotting activity of R173A mutant for the ssDNA probe (Fig. 5b), but not for ssRNA (Fig. 5c). The Y319F mutant was assayed as the negative control, unable to unknot either **dTK$_j$** or **TK$_j$** due to the loss of the active-site tyrosine residue.

To further determine the ssDNA unknotting efficiency of the R173A mutant, the concentration-dependent topological conversion assay was conducted with the ssDNA probe **dTK$_j$** (Fig. 5d). It turns out that the R173A mutant has almost identical ssDNA unknotting activity as the WT enzyme[28], both catalysing the approximately stoichiometric topological conversion of ssDNA knot within 30 min. The different activities of the WT enzyme and the two mutants are summarized in Fig. 5e. This is the first report of an amino-acid substitution in topoisomerases that affects DNA and RNA Topo activity differently. Our results imply that the region containing R173 in the domain IV, which was suggested to be important for the sequence selectivity of different Type IA DNA Topo[43], plays an important role in the enzyme's specificity to different nucleic acid substrates, for example, ssDNA, ssRNA and partially unwound dsDNA to different extents (as in sc plasmid). It is also possible that the substrate specificity can be tuned by engineering this region, and more efficient RNA Topo or RNA-specific Topo is envisioned, which can serve as a promising tool for the identification and the consequent concomitant studies of RNA topology.

**Topological inhibition of RT.** DNA topology affects the DNA-templated processes in the living cell, such as DNA replication and transcription[8]. Our previous research showed that *in vitro* the procession of various DNA polymerases can be blocked on a knotted ssDNA template[28]. Analogously, we expect that RT would also be affected by RNA topology

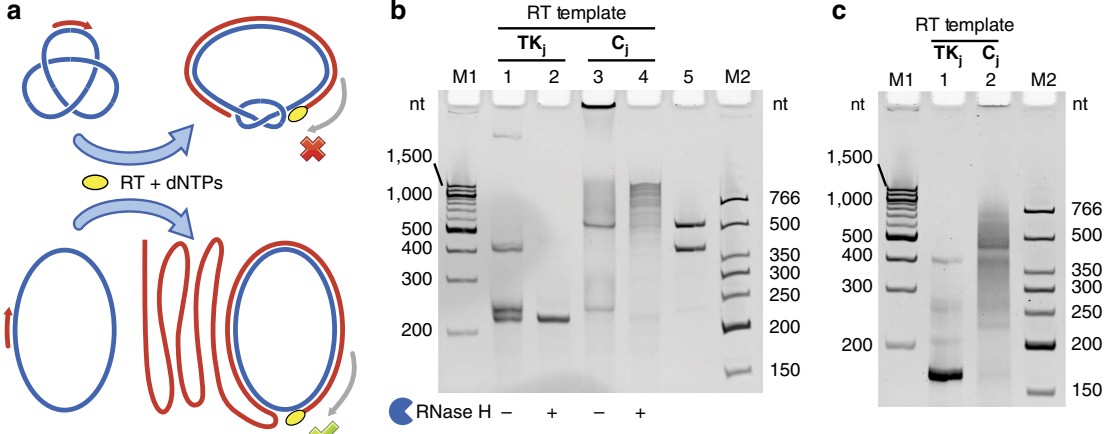

**Figure 6 | Different RT–PCR patterns resulting from RNA of different topologies.** (**a**) Schematics illustrating topological blockage of RT. On knotted ssRNA template, RT can only extend the primer to a certain point, when the RT is stalled by the force imposed by the resistance to the bending of ds RNA–DNA hybrid and the tightening of ssRNA region. On circular ssRNA template, tandem-repeat cDNA product is generated. RNA templates are shown in blue, and DNA primer and cDNA products are in red. (**b**) RT reactions of **TK$_j$** (lanes 1 and 2) and **C$_j$** (lanes 3 and 4) analysed by dPAGE. Samples in lanes 2 and 4 were treated by RNase H. Lane 5 contains **TK$_j$** and **C$_j$** as references. (**c**) PCR reactions of the cDNA products from **TK$_j$** (lane 1) and **C$_j$** (lane 2) analysed by dPAGE.

(Fig. 6a). If a knotted ssRNA is used as the RT template, the RT enzyme can extend the primer only to a certain point, when the increased free energy associated with the tightening of the diminishing ssRNA region and bending of the ds RNA–DNA hybrid region causes the enzyme to stall. As a result, only incomplete product of complementary DNA (cDNA) is generated. In contrast, on the circular ssRNA template, the RT enzyme, as a strand-displacing polymerase, could synthesize long products containing tandem repeats of the cDNA with a rolling-circle amplification fashion. To confirm this assumption, we conducted the cDNA synthesis with the ProtoScript II reverse transcriptase (NEB) on both knotted (**TK$_j$**) and circular (**C$_j$**) ssRNA templates (228 nt) and the reaction mixtures were analysed by dPAGE (Fig. 6b). As expected, only incomplete cDNA product ($\sim$210 nt) was generated from **TK$_j$**, (lanes 1 and 2), and the rolling-circle amplification product (up to $\sim$1,500 nt) was generated from **C$_j$** (lanes 3 and 4). The faint top band in lane 1 and that sticking to the well in lane 3 are probably due to the strong association of the RT enzyme with the RNA–DNA complex, even after heat denaturation and during dPAGE. After treatment with RNase H, these bands disappear (lanes 2 and 4).

The cDNA products obtained were subsequently subjected to PCR with a pair of convergent primers to amplify a 167-nt region. Figure 6c shows that the different RT–PCR patterns from RNA of different topologies. A single major band corresponding to the correct target PCR product was observed for **TK$_j$** (lane 1). However, a smear containing multiple bands was observed with **C$_j$** (lane 2) due to the tandem-repeat sequence of its cDNA. As a result, different RT–PCR patterns have been obtained for ssRNA knot and circle with the same sequence. This topology-dependent RT–PCR feature can be utilized to identify closed RNA knots from the naturally occurring circular RNAs[45]. Assays of screening other proteins for RNA Topo activity are also envisioned.

## Discussion

Synthetic DNA topology plays a prominent role in the development of DNA nanotechnology[46,47]. In the very beginning of the field, most of the earliest DNA nanostructures were actually topological targets[29,48] and the design principles were gained by building and characterizing them on the topological level. Today, the exciting field of RNA nanotechnology is beginning to emerge[49,50]. However, it is unfortunate that there was virtually no progress on RNA topological structures after Seeman's first construction of RNA knot[21]. Synthetic RNA topology will certainly catalyse further developments in this field. In fact, considering RNA's structural and functional diversity, future work is likely to yield a plethora of design strategies and practical applications of synthetic RNA topology.

Our current work, serving as a starting point, has greatly expanded the richness of RNA topological structures, including the first realization of RNA topoisomers of both positive and negative trefoil knots, and two different forms of RNA–DNA hybrid structures. The tools and methods that have been demonstrated or suggested in our work can help solve several unexplored problems associated with RNA topology. Aside from searching for naturally occurring RNA topological structures, whether or not there exists RNA-specific Topo is also a fundamental question worth pursuing. Though RNA Topo activity has been spotted in some proteins, there is to date no evidence supporting the cellular functions of these proteins as being directly related to the RNA Topo activity. Hypothetically, RNA Topos may correct the misfolded RNA structures or resolve RNA entanglements that could inhibit the normal functions. If so, this will open new opportunities for fundamental RNA biochemical and biophysical research, as well as for novel therapeutic inventions.

## Methods

**RNA preparation.** The design of the sequences followed the rule of sequence symmetry minimization[51] and was assisted by the programme CANADA[52]. Details of the sequences for each topological construct are shown in Supplementary Tables 2–5. All RNA molecules were synthesized by *in vitro* transcription using the HiScribe T7 High Yield RNA Synthesis Kit from the New England Biolabs (NEB). The corresponding DNA templates were generated by the PCR amplification of the gBlocks gene fragments from the Integrated DNA Technologies using the Q5 Hot Start High-Fidelity DNA Polymerase (NEB). To enhance the ribozyme cleavage, five thermal cycles were performed after transcription, with each cycle containing three steps: 70 °C for 10 s, 50 °C for 1 min and 37 °C for 10 min. The target RNA molecules were then purified by dPAGE. Each purified RNA molecule was treated with T4 polynucleotide kinase (NEB) in 1 × T4 DNA ligase buffer (NEB, 1 × buffer: 50 mM Tris-HCl, 10 mM MgCl$_2$, 10 mM DTT, 1 mM ATP, pH 7.5 at 25 °C) at 37 °C for 6 h to remove the 2′,3′-cyclic phosphate[31] and to

add a phosphate to 5′-hydroxyl end. After the treatment, these RNA molecules were directly used as RNA scaffolds for the topological construction.

**Topological construction.** The ssRNA topological constructs were prepared with the all-in-one protocol. It involves two steps: (1) Annealing to get the assembly complex. Equimolecular quantities (normally with a final concentration of 1 μM each) of all strands (RNA scaffolds, DNA staples and splints) were mixed in a buffer with the ultimate concentration of 1 × T4 DNA ligase buffer (by adding 10 × T4 DNA ligase buffer because the kinase-treated RNA scaffolds were in 1 × buffer) and annealed from 70 to 16 °C over 4 h. (2) Ligation to seal the nicks. To each 100 μl of reaction mixture, 4 μl of T4 RNA ligase 2 (NEB, 10 U μl$^{-1}$), 1.5 μl of 100 mM fresh DTT (NEB) and 1.5 μl of 100 mM fresh ATP (NEB) were added and incubated at 16 °C at least 16 h for the ligation. The ds RNA–DNA hybrid structures were prepared using the corresponding ssRNA knots as templates, which were prepared according to previous work[28]. The complementary RNA strands were designed as three substrands, which were annealed to the ssDNA knot templates with a ratio of complementary:template = 1.2:1. After the annealing, T4 RNA ligase 2 was used to seal the nicks in complementary RNA strands.

**dPAGE.** Gels of different concentrations were prepared using 30% acrylamide and bis-acrylamide solution (Bio-Rad, 29:1) with 7 M urea in 0.5 × TBE buffer (Bio-Rad) and run on a PROTEAN II xi cell (Bio-Rad) or a Mini-PROTEAN Tetra cell (Bio-Rad). Samples were mixed 1:1 with TBE-urea sample buffer (Bio-Rad) and heated at 70 °C for 5 min before they were loaded into the wells. Gel concentrations were carefully chosen to ensure the proper separations between different topologies as well as the references. For imaging, gels were stained with GelRed (Biotium), and images were taken by Gel Doc XR+ (Bio-Rad) imaging system and processed by software Image Lab (v.4.0.1, Bio-Rad). For purification, gels (without staining) were visualized by UV shadowing against a fluorescent thin-layer chromatography plate and bands of interest were cut. The bands were then eluted using the crush-and-soak method and the eluent was purified and concentrated on 3 K Nanosep filters (Pall). The concentration of product was determined by measuring the OD$_{260}$. Optionally, the ssRNA knots and circles can be digested by RNase R (Epicentre) after the gel extraction to remove the unavoidable cleaved linear RNA during the purification. However, RNase R digestion is not useful for the hybrid BR.

**Digestion with various nucleases.** Various nucleases were used in this work, including Nt.AlwI (NEB), Nt.BspQI (NEB), RNase H (NEB), DNase I (NEB) and RNase R (Epicentre). We used the reaction conditions for these enzymes as recommended by the providers.

**Topoisomerase assay.** The *E. coli* Topo I WT and mutant proteins were expressed and purified as described in previous study[53]. Commercial product of *E. coli* Topo I (NEB) was also tested and its RNA Topo activity was found to be slightly higher than that of the in-house prepared WT enzyme, probably due to the contamination of *E. coli* Topo III (Supplementary Fig. 8). The reaction buffer contained 1 × NEBuffer 4 (NEB, 1 × buffer: 50 mM KOAc, 20 mM Tris-acetate, 10 mM Mg(OAc)$_2$, 1 mM DTT, pH 7.9 at 25 °C) and 100 μg ml$^{-1}$ BSA. The concentration of substrates and proteins were described in the text. Reactions were quenched by phenol–chloroform extraction followed by ethanol precipitation. The reactions were then analysed by dPAGE.

**AFM imaging.** For AFM imaging of the ds RNA–DNA hybrid structures, 30 μl of 0.1 mg ml$^{-1}$ polyornithine (Sigma) solution was added to freshly cleaved mica and stand for 3 min to increase the binding to the structures before applying the samples. Then the mica was rinsed with 1 ml water and dried with compressed air. An aliquot of 5 μl of each sample (about 5 nM) in 1 × TAE-Mg buffer (11 mM MgCl$_2$, 40 mM Tris, 20 mM acetic acid, 1 mM EDTA, pH 8.0) was applied to the treated mica and stand for 1 min. Then the mica was rinsed with 1 ml water and dried with compressed air. AFM imaging was performed on a Veeco MultiMode 8 AFM in the ScanAsyst in air mode using the scanasyst-air tips (Veeco). The AFM images were processed with the software Gwyddion.

**RT–PCR.** The ProtoScript II First Strand cDNA Synthesis Kit (NEB) was used for the RT. Each reaction (20 μl) contained 100 nM ssRNA template (**TK$_j$** or **C$_j$**) and 200 nM primer (sequence is shown in Supplementary Table 6) and followed the recommended protocol. After RT, the enzyme was heat-inactivated at 80 °C for 5 min. Then the reactions were treated by RNase H. For the subsequent PCR, HotStart-IT FideliTaq DNA Polymerase (Affymetrix) was used and 1 μl of each RT mixture was added to each 50 μl of PCR reaction for amplification.

**Data availability.** Data supporting the findings of this study are available within the article and its Supplementary Information Files, and from the corresponding author on reasonable request.

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

## Acknowledgements

D.L. acknowledges the HHMI International Student Research Fellowship. This work is supported by the University of Chicago, the NSF CAREER Award (DMR-15,55,361) to Y.W. and an NIH grant (R01GM102489) to J.A.P.

## Author contributions

D.L. and Y.W. conceived the project. D.L. designed the experiments, performed the research and analysed the data. Y.S. and G.C. assisted the biochemical assays. Y.-C.T.-D. contributed the Topo I enzymes. Y.-C.T.-D. and J.A.P. advised the assays. All authors discussed the results and wrote the paper.

## Additional information

**Competing interests:** The authors declare no competing financial interests.

