## [Peer Review File · Nature Communications]

Reviewer #1 (Remarks to the Author):

In this manuscript, the authors demonstrate their ability to generate non-trivial structures of closed RNA loops exhibiting desired topologies, adapting a technique they recently developed for DNA. They present two strategies, using either DNA / RNA hybrid four way junctions or by 'stapling' RNA's with a set of DNA templates that are then removed. Strikingly, these constructs allow authors to confirm that DNA Topoisomerase I has activity on RNA knots (recently determined independently: Ahmad, M. et al. *Nucleic Acids Res.* 2016), and also show that a well studied point mutation, R173A abolishes unknotting activity in RNA. Lastly, also using RNA trefoil knots, they demonstrated that RNA Reverse Transcriptase activity is inhibited, similar to a DNA effect with DNA polymerase I that they demonstrated in a previous publication. Overall this work not only represents a step forward in nucleic acid design of structural topologies but also contains new constructs with nice biochemical assays for testing RNA topoisomerase activity which will be useful for possibly new discoveries of RNA topoisomerases. We recommend that this manuscript be accepted for publication after addressing some minor issues:

1. Reuse of figures from previous paper (Liu, D., Chen, G., Akhter, U., Cronin, T. M. & Weizmann, Y.). Figures are almost identical from Figure 1C compared to Figure 1A from this paper. Similarly figure 3A is almost identical 4B from this paper. The only difference seems to be that the cartoon was re-rendered with a slightly different style.
2. There is a lot of colloquial language is present in this paper including which may not be suitable for Nature Communications:
"In fact, the biological importance of DNA topology can partly be reflected by the diversity of DNA topoisomerases (DNA Topos), which, as "the magicians of the DNA world", are neat tools of Nature to creatively solve the topological puzzles of DNA in the living cell."
"In the field of chemical topology, molecular BR is considered an Everest. Like the real-world Mount Everest, the fact that it has been conquered once and again"
3. In figure 4 there appears to be an extra band running quickly in most Topo I activity gels (c, d, e) what is this?
4. In figure 3, other than hybrid knots and hybrid circles that which have corresponding AFM images how were the running positions of the other topologies determined on these gels other than DNase and RNase activity?

Reviewer #2 (Remarks to the Author):

This paper describes the synthesis and analysis of RNA topological structures, especially RNA-containing knots, by novel methodologies, and their analysis by gel electrophoresis, AFM and topoisomerase assay. The authors demonstrate how it is possible to generate and analyze a range of RNA topological structures. Although this is fascinating stuff I am really not sure that this work is well-suited to this journal. The following issues should be brought to the authors' attention:

1. My main concern about this ms is that it is about RNA structures looking for a function. Currently it is not at all clear that such topological forms of RNA exist in nature. On this basis I feel the work would be better suited to a more specialized journal.
2. In the Introduction the authors say the DNA topo III is an RNA topoisomerase. I don't think this is actually true; it is a DNA topoisomerase that can also catalyze RNA topoisomerase reactions, in common with other topoisomerases (see ref 24).
3. The authors assess the ability of DNA topo I to catalyze RNA topoisomerase reactions. This struck me as an odd choice. In the light of ref 24 it would have been useful to survey other enzymes.

4. I found the figures to be very crowded and complex; they need to be expanded and clarified.

Reviewer #3 (Remarks to the Author):

Liu et al. presents an extensive study on synthesizing and investigating topological RNA structures. Eventhough several interesting properties are observed the study has several issues concerning experimental design and conclusions that are not well supported by the data. Due to these problems we cannot recommend publication at this point.

Major concerns:

1) One of the main problems is that the authors claim that their probe (TK_j) is better than the former Seeman probe (C_h) to assay Topo I activity. This is however not proven by the data shown in Fig. 4c,d because of the difference in topology between the probes used. In Fig. 4c Topo I releases 3 negative knots from the probe TK_j and forms a relaxed circle - an activity which is similar to the natural activity of the enzyme Topo I. However, in Fig. 4d they expect Topo I to add 3 negative knots to a relaxed circle (probe C_h), making the two assays uncomparable.

2) Changing of several factors at once: We have seen this many places for example in Fig. 4gh, where the unknotting activity is assayed on two different substrates and the results compared. However, the nature of the two substrates are different. In Fig. 4g the substrate is DNA with positive knots, while in Fig. 4h the substrate is RNA with negative knots.

3) Lack of proper controls in gel assays. In e.g. Fig. 4e. they use TK_j as a control, but that is not appropriate for this experiment. Instead they should have used substrate (C2_h) and product (TK2_h) as controls along with GK_h.

4) Insufficient annotation of band and lanes on gels. Example is Fig. 4d where a lower band changes as a function of Topo I concentration, but is not annotated nor taken into consideration for the data analysis.

First of all, we would like to express our sincere gratitude to all the reviewers for their careful efforts to review our manuscript. We are very pleased and encouraged to learn that all the reviewers are showing interest in our work. There are a number of suggestions that have helped significantly strengthen our manuscript. Also, a few points of misunderstanding by our reviewers showed us where the manuscript should be written more clearly. Following is our point-by-point response to address the concerns raised by our reviewers.

Point-by-point response

Reviewer #1 (Remarks to the Author):

“In this manuscript, the authors demonstrate their ability to generate non-trivial structures of closed RNA loops exhibiting desired topologies, adapting a technique they recently developed for DNA. They present two strategies, using either DNA / RNA hybrid four way junctions or by ‘stapling’ RNA’s with a set of DNA templates that are then removed. Strikingly, these constructs allow authors to confirm that DNA Topoisomerase I has activity on RNA knots (recently determined independently: Ahmad, M. et al. Nucleic Acids Res. 2016), and also show that a well studied point mutation, R173A abolishes unknotting activity in RNA. Lastly, also using RNA trefoil knots, they demonstrated that RNA Reverse Transcriptase activity is inhibited, similar to a DNA effect with DNA polymerase that they demonstrated in a previous publication. Overall this work not only represents a step forward in nucleic acid design of structural topologies but also contains new constructs with nice biochemical assays for testing RNA topoisomerase activity which will be useful for possibly new discoveries of RNA topoisomerases. We recommend that this manuscript be accepted for publication after addressing some minor issues:

1. Reuse of figures from previous paper (Liu, D., Chen, G., Akhter, U., Cronin, T. M. & Weizmann, Y.). Figures are almost identical from Figure 1C compared to Figure 1A from this paper. Similarly figure 3A is almost identical 4B from this paper. The only difference seems to be that the cartoon was re-rendered with a slightly different style.”

(1) We thank Review #1 for the high evaluation of the new strategies we have presented in the current work of synthesizing RNA-containing topological structures and their application in several biochemical assays. We are very sorry that some illustrations in the current manuscript look similar to our previous DNA topology paper. However, they are not identical. **Figure 1c** in the current manuscript is the RNA-DNA hybrid four-way junction, so the helices presented are A-form. In contrast, **Figure 1a** in our previous paper is DNA four-way junction, in which the helices are B-form. In the revised version of the current manuscript, we further change the color of the illustrations in **Figure 1a** and **1c**: the RNA strands in **Figure 1a** are colored in orange; those in **Figure 1c** are colored in blue. These changes make the coloring more consistent throughout the manuscript: RNA structures colored in orange are those containing strong

intramolecular base-pairings; RNA structures colored in blue are those free of strong intramolecular base-pairings.

The 3D model in **Figure 3a** of the current manuscript is not the same as that in **Figure 4b** of the previous paper. Both models are accurately built based on our geometrical analysis. The model in **Figure 3a** of the current manuscript contains the double-stranded RNA-DNA hybrid helices (A-form) and the inner/outer helices are 42/34 bp. In contrast, the model in **Figure 4b** of the previous paper contains the double-stranded DNA helices (B-form) and the inner/outer helices are 40/32 bp. Therefore, these two 3D models are not the same model with different rendering.

“2. There is a lot of colloquial language present in this paper including which may not be suitable for Nature Communications:

“In fact, the biological importance of DNA topology can partly be reflected by the diversity of DNA topoisomerases (DNA Topos), which, as “the magicians of the DNA world”, are neat tools of Nature to creatively solve the topological puzzles of DNA in the living cell.”

“In the field of chemical topology, molecular BR is considered an Everest. Like the real-world Mount Everest, the fact that it has been conquered once and again””

(2) Thanks for pointing this out. These two sentences are now revised to remove the colloquial expressions and are highlighted in the manuscript file.

“3. In figure 4 there appears to be an extra band running quickly in most Topo I activity gels (c, d, e) what is this?”

(3) The same concern on the faster-migrating bands is also raised by Review #3. These bands are the linear break-down (nicked) species from the closed RNA structures. The breaking down of RNA is unavoidable during the RNA digestion, handling and storage due to the much poorer stability of RNA comparing to DNA. In the revised version of the manuscript, the bands corresponding to the linear break-down species in all the relevant gels are clearly annotated.

“4. In figure 3, other than hybrid knots and hybrid circles that which have corresponding AFM images, how were the running positions of the other topologies determined on these gels other than DNase and RNase activity?”

(4) We have designed our experiments very carefully to provide the appropriate references for the gel electrophoresis assays. For this purpose, we have designed the length of the junction-based RNA structures (228 nt) to be the same as the RNA components in the hybrid structures, which are also twice the length of the monomeric helix-based RNA structures (114 nt). In **Figure 3c**, lanes 7 and 8 contain the ssDNA knot and circle, respectively. The ssDNA knot and circle are prepared using the junction-based method and are the templates for the synthesis of the hybrid

knot and circle. They also serve as the references for the samples digested with RNase H (lanes 3 and 6). Lanes 9 and 10 contain the references of ssRNA knot $TK_j(+)$ and circle C_j (also shown in **Figure 2c**), which are of the same size (228 nt) with the RNA components in the hybrid species, but a different sequence. They also serve as the references for the samples digested with DNase I (lanes 2 and 5).

More generally, the electrophoretic mobility of the topological structures are determined by both comparing the mobility of related topological structures and enzymatic digestions. In most cases, the electrophoretic mobility of different structures of the same total chain length in denaturing gel follows linear > knot > circle (in low-concentration gels the knot of short length may catch up with the linear). The conventional molecular size marker can only indicate the size of the linear species. The linear and circular species must be prepared beforehand in order to identify the knot. The circle can be generated by the ligation in the absence of staple strands and the linear species can be generated by further removing one splint strands during the ligation. Three kinds of enzymes are usually useful in the further confirmation of topological structures. (1) Exonucleases. Exonucleases can degrade all the linear species and only the closed species can survive. As for RNA, the exonuclease used is RNase R, as is shown in **Figure 2c**. (2) Restriction enzymes. For some complex topological structures, we have designed restriction sites in the sequence so that the topology can be further proved by selectively cleave any components in a certain structure. An example in case is the hybrid BR shown in **Figure 2e**. (3) Topoisomerase. Topoisomerase can resolve the structure to its most thermodynamically stable state and can further provide evidence of the topology. In our paper, the topoisomerase assays with the RNA knot not only demonstrate the RNA topoisomerase activity of the enzyme, but also validate the topology of the RNA knot.

Reviewer #2 (Remarks to the Author):

“This paper describes the synthesis and analysis of RNA topological structures, especially RNA-containing knots, by novel methodologies, and their analysis by gel electrophoresis, AFM and topoisomerase assay. The authors demonstrate how it is possible to generate and analyze a range of RNA topological structures. Although this is fascinating stuff, I am really not sure that this work is well-suited to this journal. The following issues should be brought to the authors’ attention:

1. My main concern about this ms is that it is about RNA structures looking for a function. Currently it is not at all clear that such topological forms of RNA exist in nature. On this basis I feel the work would be better suited to a more specialized journal.”

(1) We appreciate Reviewer #2 for the recognition of the synthesis and analysis described in our work. As for the first concern raised by the reviewer, we would like to point out that RNA topology is a largely unexplored area in the current RNA research. A main reason accounting for

this fact is that there is no available material for the study of RNA topology. As we know, the supercoiled DNA has been a very handy material for investigating DNA topology and DNA topoisomerases. However, we do not have such luxury in the case of RNA and this is why we take so many efforts in the synthesis and analysis of the RNA topological structures. In fact, the topological property is one of the most elementary properties for all sufficiently long molecules. We believe that our understanding of RNA and the way it is processed in cell could substantially profit from a better knowledge of this property.

Furthermore, even though no knotted RNA has been validated yet in nature, there is evidence that RNA may adopt complex topology *in vivo* and that topology can render the RNA functions. As has been mentioned in our manuscript, several RNA pseudoknots are likely to be real knots based on their sequences. As for the functions, in a 2014 Science paper (Science 344(6181):307-310 (2014)), Kieft and co-workers reported the structure of the subgenomic flaviviral RNA (sfRNA). In this structure, the 5' end is threaded through a ring-like structure formed by two helices, creating a topological barrier for the 5' to 3' host cell exonuclease Xrn1. This effect is analogous to the topological inhibition of reverse transcription presented in our paper. Even though this sfRNA is still not a real knot by strict mathematical definition, the unique topology of its 3D folding contributes to its resistance to Xrn1.

Considering the potentially vast opportunities in the research of RNA topology, we hope that this paper can make more researchers aware of this area by taking advantage of the high impact and the wide readership of *Nature Communications*.

“2. In the Introduction the authors say the DNA topo III is an RNA topoisomerase. I don't think this is actually true; it is a DNA topoisomerases that can also catalyze RNA topoisomerase reactions, in common with other topois (see ref 24).”

(2) We had been very careful in using the term “RNA topoisomerase”. *E. coli* DNA Topo III was the only enzyme that we called an RNA topoisomerase in our original manuscript and all the other enzymes which can catalyze the RNA topological conversion were called to have the RNA topoisomerase activity. The reason we called *E. coli* DNA Topo III an RNA topoisomerase is that it was called so in Seeman's PNAS paper published in 1996 (Ref#18, and the title of this paper is “*An RNA topoisomerase*”). I think in this scenario the term “RNA topoisomerase” is used if the enzyme has RNA topoisomerase activity (not necessarily RNA-specific) and this convention has been accepted among the researchers in this area (in fact, the term “RNA topoisomerase” also appears in the titles of Ref#23 and Ref#24).

For the sake of better accuracy of expression, we stop saying *E. coli* DNA Topo III an “RNA topoisomerase” but rather saying it possessing “RNA topoisomerase activity” in the revised version of the manuscript. The revised sentences are highlighted in the manuscript text file.

“3. The authors assess the ability of DNA topo I to catalyze RNA topoisomerase reactions. This struck me as an odd choice. In the light of ref 24 it would have been useful to survey other enzymes.”

(3) For us, *E. coli* DNA Topo I was a very appropriate starting point to probe the RNA topoisomerase activity when we were designing our experiments. Firstly, a lot of DNA-processing enzymes can also catalyze the similar reactions with RNA substrates. *E. coli* DNA Topo I is a Type IA topoisomerase, which acts on ssDNA. Therefore, this enzyme is very likely to possess the topoisomerase activity on ssRNA and can serve as a model to test our constructed ssRNA knot as a RNA topoisomerase probe. Secondly, *E. coli* DNA Topo I was reported to lack the RNA topoisomerase activity (Ref#18) using Seeman’s helix-based probe. We assume that the negative result might be due to the intrinsic intramolecular base-pairings within the structure, which limits the sensitivity of the helix-based probe. Eventually, we independently discovered the relatively low RNA topoisomerase activity of *E. coli* DNA Topo I and validated our assumption that our junction-based probe is indeed more sensitive than the previous helix-based probe in detecting RNA topoisomerase activity. Thirdly, Topo I is present in every bacterium, so the RNA topoisomerase activity of Topo I would be relevant if RNA topological problems need to be resolved *in vivo*.

Having proved that our junction-based RNA probe is more sensitive comparing with the helix-based probe using *E. coli* DNA Topo I as a model, we reasonably anticipate that our probe can be utilized to assay other enzymes for the RNA topoisomerase activity. Indeed, searching for new RNA topoisomerases is a future direction in our lab, but this will be out of the scope of our current work.

“4. I found the figures to be very crowded and complex; they need to be expanded and clarified.”

(4) We have divided **Figure 4** into **Figures 4** and **5** in the revised version of the manuscript. Now, the maximum number of panels in all of our figures is five. The panels of the figures are also rearranged for better clarity.

Reviewer #3 (Remarks to the Author):

“Liu et al. presents an extensive study on synthesizing and investigating topological RNA structures. Even though several interesting properties are observed, the study has several issues concerning experimental design and conclusions that are not well supported by the data. Due to these problems we cannot recommend publication at this point.

Major concerns:

1) One of the main problems is that the authors claim that their probe (TK j) is better than the former Seeman probe (C h) to assay Topo I activity. This is however not proven by the data

shown in Fig. 4c,d because of the difference in topology between the probes used. In Fig. 4c Topo I releases 3 negative knots from the probe TK_j and forms a relaxed circle - an activity which is similar to the natural activity of the enzyme Topo I. However, in Fig. 4d they expect Topo I to add 3 negative knots to a relaxed circle (probe C_h), making the two assays uncomparable.”

(1) Concerns given by Reviewer #3 regarding the experimental design of our work are very constructive. As for the first concern about comparing our probe TK_j with Seeman’s probe C_h, we would like to respectively justify that these two probes can be compared and our junction-based probe is indeed more sensitive than the previous helix-based probe. This is our logic: if probe A can detect the topoisomerase activity at a lower enzyme-to-probe molar ratio or enzyme-to-nucleotide ratio than that of probe B, we can confidently claim that probe A is more sensitive than B. It is really not necessary that these two probes should be of the same topology. In our case, the topological conversion of TK_j starts to be observable in the lane 1 of **Figure 4c** (where the enzyme-to-probe ratio is 1:2 and the enzyme-to-nucleotide ratio is 1:456). In contrast, the conversion of C_h can be observed in the lane 3 of **Figure 4d** (where the enzyme-to-probe ratio is 2:1 and the enzyme-to-nucleotide ratio is 1:57). Therefore, it is valid to claim that our junction-based probe TK_j is more sensitive than the helix-based probe C_h in detecting topoisomerase activity.

Here, we also want to explain that as for the helix-based structures, the circle C_h, instead of the knot TK_h, should be used as the topoisomerase probe. The use of nucleic acid topological structure as topoisomerase probe was first suggested and demonstrated by Seeman (Biochemistry 34 (2): 673–682 (1995)). In order to be the topoisomerase probe, the topological structure should be in a thermodynamically higher energy state. In other word, it should be stressed and be prone to transform into a different topology in the presence of topoisomerase. As shown in the following figure, C_h is the stressed topological structure because the topology of the circle cannot accommodate the formation of the two 2-turn helices without forming extra crossovers (or nodes) as highlighted by the red oval. This has also been explained in **Figure 4b** and its accompanying text in the manuscript. The fact that C_h is stressed can also be reflected by the necessity of adding an extra block strand as well as a longer splint strand in its synthesis to suppress the formation of the helices (**Supplementary Figure 5**).

In fact, the helix-based circle (C_h), instead of the knot (TK_h), is virtually used as the probe to detect the RNA topoisomerase activity in recent researches (Ref#23 and Ref#24). The reviewer may also need to have a look at Fig. 3 of Seeman's 1996 PNAS paper (Ref#18). The conversion of circle to knot is more favorable over the conversion from knot to circle (even though the latter is conducted in the presence of a 40-nt complementary strand to suppress the knot formation). We further conducted a similar experiment with *E. coli* DNA Topo I and the result (**Supplementary Figure 7**) also demonstrates that the circle (C_h) should be used as the probe.

Furthermore, the expressions of “*releases 3 negative knots*” and “*add 3 negative knots*” in the reviewer's comments are not very accurate in depicting the action of Topo I. It should be noted that the number of nodes is not an invariant of a knot. Instead, the minimum number of nodes is a knot invariant. However, the minimum projection (a projection with the minimum number of nodes) is not always the most appropriate in presenting the real conformation of a topological structure, especially for the helix-based structures in native condition. Examples of this point can be found in the above illustrations of C_h and TK_h . The number of nodes is 5 and 4 for C_h and TK_h , respectively, rather than 0 and 3 (their respective minimum numbers of nodes). Therefore, “catalyzing the strand passage reaction” can be a more proper expression in depicting the action of Topo I.

“2) Changing of several factors at once: We have seen this many places for example in Fig. 4gh, where the unknotting activity is assayed on two different substrates and the results compared. However, the nature of the two substrates are different. In Fig. 4g the substrate is DNA with positive knots, while in Fig. 4h the substrate is RNA with negative knots.”

(2) Using DNA and RNA knots with different handednesses seems to be confusing. To dispel this confusion, we have prepared the negative DNA knot and replace **Figures 4g** and **4i** with the results obtained using the negative DNA knot (**Figures 5b** and **5d** in the revised version of manuscript). (In **Figures 5d**, we also changed the concentration range of enzyme from the previous 20-to-320 nM to the current 10-to-160 nM in lanes 1-5 because the knot is completely converted to circle at above 160 nM of enzyme.) Because the topological structures prepared with our junction-based method have no intrinsic strong base-pairings, it really does not matter whether positive or negative knot is used in the assay and our results (previous and current) indeed prove so. Therefore, our previous conclusions are not changed.

However, we still would like to justify that our initial experiment design should have no problem. We are not comparing **Figures 4g** and **4h** (**Figures 5b** and **5c** in the revised version of manuscript). What we are comparing is the activities of the WT and mutated Topo I against three different substrates (ssDNA and ssRNA in the current work, and supercoiled plasmid in previous research by other groups) in order to summarize their substrate selectivity tabulated in **Figures 5e**. The reason we chose the positive DNA knot as substrate in our previous experiment is that the positive knot was used in the topoisomerase assays in our previous research (REF#28) and we wanted to keep it consistent. Because the RNA knot prepared in the helix-based method has

negative nodes, we chose the negative RNA knot in the topoisomerase assay in the current work. Again, both positive and negative knots prepared with our junction-based method are in a more stressed state compared to the circle and, therefore, the results and conclusions should not be changed by the handednesses of the knots.

“3) Lack of proper controls in gel assays. In e.g. Fig. 4e. they use TK_j as a control, but that is not appropriate for this experiment. Instead they should have used substrate (C2_h) and product (TK2_h) as controls along with GK_h.”

(3) **Figure 4e** presents our extra effort to improve the sensitivity of the helix-based probe by introducing more topological stress. In **Figure 4e**, TK_j serves as a molecular size marker, rather than as a control. In this gel, we have already included the substrate C2_h (dimeric circle) and the ultimate product GK_h (dimeric granny knot) as the control. We determine the intermediate product to be the dimeric trefoil knot TK2_h by combining the fact that in the digested samples only one band appears between GK_h and C2_h, and the fact that the topological conversion from C2_h to GK_h has to go via the trefoil knot intermediate. Admittedly, it would be better if we have the intermediate TK2_h as control. However, the dimeric nature of its sequence makes it very difficult to prepare TK2_h with sufficient yield. (The reviewer may want to have a look at **Supplementary Figure 5** to learn about our strategy to synthesize the helix-based structures in this work.) Therefore, we choose TK_j as the marker or reference because it is of the same length and same topology with TK2_h. (The reviewer may also need to read our **Response (4)** to Review #1 to learn about how we design the molecular size of the structures in order to have proper references for the electrophoresis experiments.) The fact that TK2_h migrates slightly faster than TK_j in the dPAGE is likely due to the strong intrastrand base-pairings in TK2_h as was also observed previously, and we have already pointed this out in our manuscript.

“4) Insufficient annotation of band and lanes on gels. Example is Fig. 4d where a lower band changes as a function of Topo I concentration, but is not annotated nor taken into consideration for the data analysis.”

(4) The faster-migrating bands in the gels are the linear break-down (nicked) species from the closed RNA structures. We have addressed this concern in our **Response (3)** to Review #1. In the reactions containing very high concentration of Topo I, higher levels of RNA break-down products are observed most probably due to the enzyme's failure in ligation, which is the final step of the strand-passage catalysis. Though the presence of the break-down products prevents us from quantifying the conversion very accurately, we have indeed taken this factor in our data analysis. We estimate that the RNA Topo activity of Topo I is about 1/15-1/12 of its DNA Topo activity. The lower limit is gotten if the break-down products are solely from the unconverted substrate, while the upper is gotten if the break-down products are from the substrate and product proportionally. The real value should fall in between.

Thanks to the careful efforts of the reviewers, our revised manuscript has now been substantially improved. We anticipate that our work will bring in more refreshing inspirations to the scientific community by attracting more researchers' attention to this exciting but unexplored area of RNA topology.

Best regards,

Yossi

Reviewer #1 (Remarks to the Author):

The authors have addressed our concerns to our satisfaction.

Reviewer #2 (Remarks to the Author):

This is a revised version of a recently reviewed ms. In relation to my comments in my original review I offer the following updates:

1. Relating to my concern about the paper being about 'RNA structures looking for a function', the authors have addressed this comment, and, although I don't necessarily agree with their response, it is nonetheless an acceptable viewpoint.
2. Relating to my comment about topo III being identified as an RNA topoisomerase, the authors have satisfactorily addressed this.
3. Regarding using E. coli topo I as their test enzyme, I still feel they should have tested other, more relevant, enzymes, but I can accept that this is something that can be done in the future.
4. The figures are somewhat improved.

Also:

5. In response to reviewer 1, the authors have tried to remove some of the colloquial language. Referring to the sentence at the end of the first paragraph of the main text, I think it is not the 'magicians' comment that is at fault (this is a direct quote from JC Wang) but the remainder of the sentence about topoisomerases being 'neat tools' etc. Can I suggest something more prosaic: DNA topoisomerases (DNA Topoisomerase) evolved to solve topological problems in DNA.

Reviewer #3 (Remarks to the Author):

Liu et al. have made a revision of their manuscript where they have answered the concerns of the reviewers and improved the presentation of their study significantly. The improved text and figures allow the reader to appreciate the results of the study better. I can now recommend publication, but since I am still left with a few questions after rereading the manuscript I think that the authors should do a few more updates to help the reader.

Clarifications:

- 1) Please report the sensitivity value for the TK_j probe in more detail. Right now you just relate it to the DNA probe from a former study. Since you are comparing with the C_h probe then report the sensitivity in similar terms. As far as I can gather you have a conversion of 7-8% for the TK_j probe as compared to 3% of the C_h probe. In your response to the reviewers you also mention that the kinetics is different. Please write this in the manuscript text, so that the reader is provided with a clear understanding of the improvement.
- 2) Why is Fig. 2e, lane e not just one band? Please explain why your purification procedure still has more bands on the gel.
- 3) Fig. S6 a and b: Why does the mobility of C_h and TK_h change between the two gels. Did the percentage of the PAGE gel change to induce different mobilities? Please explain.
- 4) In Fig. S7, lane 2: It is very difficult to see the product of the reaction. Could you provide a gel image besides the gel, where you increase the contrast to allow the reader to see the band you are talking about?
- 5) I appreciate that you have annotated bands better in the main figures. However, many bands in the supplementary material are still not annotated. Could you provide a bit more information about

what all the species are or write that they are unknown.

Corrections:

- 1) You write that you shorten the TK_j(-) abbreviation to "clarify". I think you mean "simplify".
- 2) In Fig. 2 caption: Change "E. coli." to "E. coli".
- 3) In Fig. 2 caption: Change "for and" to "for".
4. I found the figures to be very crowded and complex; they need to be expanded and clarified.

Reviewer #3 (Remarks to the Author):

Liu et al. presents an extensive study on synthesizing and investigating topological RNA structures. Eventhough several interesting properties are observed the study has several issues concerning experimental design and conclusions that are not well supported by the data. Due to these problems we cannot recommend publication at this point.

Major concerns:

- 1) One of the main problems is that the authors claim that their probe (TK_j) is better than the former Seeman probe (C_h) to assay Topo I activity. This is however not proven by the data shown in Fig. 4c,d because of the difference in topology between the probes used. In Fig. 4c Topo I releases 3 negative knots from the probe TK_j and forms a relaxed circle - an activity which is similar to the natural activity of the enzyme Topo I. However, in Fig. 4d they expect Topo I to add 3 negative knots to a relaxed circle (probe C_h), making the two assays uncomparable.
- 2) Changing of several factors at once: We have seen this many places for example in Fig. 4gh, where the unknotting activity is assayed on two different substrates and the results compared. However, the nature of the two substrates are different. In Fig. 4g the substrate is DNA with positive knots, while in Fig. 4h the substrate is RNA with negative knots.
- 3) Lack of proper controls in gel assays. In e.g. Fig. 4e. they use TK_j as a control, but that is not appropriate for this experiment. Instead they should have used substrate (C2_h) and product (TK2_h) as controls along with GK_h.
- 4) Insufficient annotation of band and lanes on gels. Example is Fig. 4d where a lower band changes as a function of Topo I concentration, but is not annotated nor taken into consideration for the data analysis.

We would like to thank the reviewers again for their helpful suggestions. We are pleased to learn that our revised manuscript and the point-by-point response have satisfactorily addressed all the major concerns raised by the reviewers. We hope that the remaining concerns could be addressed after this round of revision.

Point-by-point response

Reviewer #1 (Remarks to the Author):

“The authors have addressed our concerns to our satisfaction.”

We want to thank Review #1 again for the support and the efforts to help improve our paper.

Reviewer #2 (Remarks to the Author):

“This is a revised version of a recently reviewed ms. In relation to my comments in my original review I offer the following updates:

1. Relating to my concern about the paper being about ‘RNA structures looking for a function’, the authors have addressed this comment, and, although I don’t necessarily agree with their response, it is nonetheless an acceptable viewpoint.”

(1) We appreciate the constructive discussion from Reviewer #2 regarding the function of the RNA topological structures, which, admittedly, is still unclear at the moment. We believe that our current work can serve as a starting point to solve this problem in the future.

“2. Relating to my comment about topo III being identified as an RNA topoisomerase, the authors have satisfactorily addressed this.”

(2) Thanks!

“3. Regarding using E. coli topo I as their test enzyme, I still feel they should have tested other, more relevant, enzymes, but I can accept that this is something that can be done in the future.”

(3) Thanks for understanding! Surveying other enzymes for RNA Topo activity is a follow-up of the current work, and will be given high priority in our future research.

“4. The figures are somewhat improved.”

(4) Thanks!

“Also:

5. In response to reviewer 1, the authors have tried to remove some of the colloquial language. Referring to the sentence at the end of the first paragraph of the main text, I think it is not the ‘magicians’ comment that is at fault (this is a direct quote from JC Wang) but the remainder of the sentence about topoisomerases being ‘neat tools’ etc. Can I suggest something more prosaic:

DNA topoisomerases (DNA Topoisomerase)12-14, enzyme evolved to solve topological problems in DNA.”

(5) Thanks for the suggestion on how to phrasing this sentence more properly. We have revised this sentence accordingly.

Reviewer #3 (Remarks to the Author):

“Liu et al. have made a revision of their manuscript where they have answered the concerns of the reviewers and improved the presentation of their study significantly. The improved text and figures allow the reader to appreciate the results of the study better. I can now recommend publication, but since I am still left with a few questions after rereading the manuscript I think that the authors should do a few more updates to help the reader.

Clarifications:

1) Please report the sensitivity value for the TK_j probe in more detail. Right now you just relate it to the DNA probe from a former study. Since you are comparing with the C_h probe then report the sensitivity in similar terms. As far as I can gather you have a conversion of 7-8% for the TK_j probe as compared to 3% of the C_h probe. In your response to the reviewers you also mention that the kinetics is different. Please write this in the manuscript text, so that the reader is provided with a clear understanding of the improvement.”

(1) We are pleased that Reviewer #3 are satisfactory with our revision and now recommends publication of our work. We will try our best to present our manuscript to help the readers better appreciate our results.

The conversion of the TK_j to C_j is ~28%-35% in the lane 4 of the gel in **Figure 4c**. This information is now added to the revised manuscript. Because the enzyme-to-probe ratio in lane 4 is 4:1, we estimated that the RNA Topo activity of Topo I is ~1/15-1/12 of the DNA Topo activity (7-8% mentioned in Reviewer #3’s comment should be Topo I’s RNA Topo activity compared to its DNA Topo activity, but not the conversion of the TK_j probe -- sorry for this confusion). As for the explanation of the mechanism of helix-based probes, we have included this information (including the illustration figures we have prepared in our last response letter) in

Supplementary Figure 7 in the revised manuscript, which is explicitly referred to from the main text.

“2) Why is Fig. 2e, lane e not just one band? Please explain why your purification procedure still has more bands on the gel.”

(2) During the purification of the hybrid BR, the breaking down of the 95-nt circular RNA component is unavoidable, and consequently a portion of BR falls apart. The other bands appearing in lane 1 correspond to the 95-nt RNA linear break-down product, and the 105- and 116-nt DNA circles. We have included this information in the caption of this figure.

“3) Fig. S6 a and b: Why does the mobility of C_h and TK_h change between the two gels. Did the percentage of the PAGE gel change to induce different mobilities? Please explain.”

(3) Yes, the electrophoretic mobility of the circular and knotted species (relative to the linear species and the size markers) is different in different concentrations of gels. This is reflected in the Ferguson plots (as can be seen in Seeman’s paper, REF #21), in which the logarithm of the electrophoretic mobility (M) is plotted as a function of the gel concentration (T): $\log(M) = \log(M_0) - K_R T$. The extrapolated intercept M_0 is the free electrophoretic mobility, which is related to the net surface charge density. The slope K_R is the retardation coefficient, which is related to the apparent molecular size. Both knot and circle have larger K_R than the linear species, and therefore, their mobility increases more comparatively when decreasing the gel concentration. In our experiments, we have used different gel concentrations to better separate the bands we are interested in certain experiments. In **Supplementary Figure 7a-c**, the concentrations of gels are 8%, 10% and 7%, respectively. Taking **TK_h** as an example, it runs almost the same as **L114** in 8% gel (**Supplementary Figure 7a**), slower than **L114** in 10% gel (**Supplementary Figure 7b**), and faster than **L114** in 7% gel (**Supplementary Figure 7c**). We have included this discussion in the caption of **Supplementary Figure 7**.

“4) In Fig. S7, lane 2: It is very difficult to see the product of the reaction. Could you provide a gel image besides the gel, where you increase the contrast to allow the reader to see the band you are talking about?”

(4) We have added **Supplementary Figure 7d** presented with a different contrast level to the original gel image (**Supplementary Figure 7c**) in order to show the very faint band of product **TK_h** in lane 2.

“5) I appreciate that you have annotated bands better in the main figures. However, many bands in the supplementary material are still not annotated. Could you provide a bit more information about what all the species are or write that they are unknown.”

(5) Thanks for the comments. In the revised manuscript, we have annotated most of the major bands in the gel images in the supplementary figures. For clarity, we have also added explanatory discussions for some of the unannotated bands in the figure captions.

“Corrections:

1) You write that you shorten the TK j(-) abbreviation to "clarify". I think you mean "simplify".

2) In Fig. 2 caption: Change "E. coli." to "E. coli".

3) In Fig. 2 caption: Change "for and" to "for".”

(6) Thanks a lot for pointing out these errors. Now, corrections have been made in the revised manuscript accordingly. We have also checked the manuscript very thoroughly to correct any typos we could find.